# Modified Supergravity Phenomenology in Gravitational Waves Era

**Andrea Addazi [1,2,*] and Qingyu Gan [1]**

1   Center for Theoretical Physics, College of Physics, Sichuan University, Chengdu 610065, China;
    gqy@stu.scu.edu.cn
2   Laboratori Nazionali di Frascati INFN, Via Enrico Fermi 54, I-00044 Frascati, Italy
*   Correspondence: addazi@scu.edu.cn

**Abstract:** We discuss phenomenological aspects of modified supergravity (MSG) in gravitational wave (GW) physics. MSG naturally provides double inflation and primordial black holes (PBHs) as cold dark matter. Intriguingly, MSG predicts a large amplification of the scalar and tensor perturbation power spectrum, generating a secondary GW stochastic background which can be tested in space-based interferometers.

**Keywords:** modified gravity; supergravity; primordial black holes

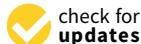



## 1. Introduction

The phenomenology of modified supergravity (MSG) is one of the next frontiers as researchers move toward a quantum gravity bottom-up approach. After the discovery of gravitational waves [1,2], a natural arena for MSG tests is multi-messenger physics. Supergravity is a key element for superstring theory, which in turn predicts higher derivative $\alpha'$ or (D-brane or worldsheet) instantonic curvature corrections. Thus, MSG can be considered as an effective (super)field theory in the low energy limit of superstring theory, eventually waiting for a better understanding of M-theory in all of its whole complexity. On the other hand, an explanation of cold dark matter (CDM) remains elusive to us. One of the most attractive ideas is that CDM is composed by primordial black holes (PBHs) (see e.g., [3–5]). As it is known, no astrophysical mechanisms can explain such a copious amount of black holes (BHs) as CDM. That is why if CDM was composed of BHs, then they would be produced by a new physics mechanism beyond the standard model (SM) in the early universe [3–5].

In the cosmological perturbation theory, the scalar and tensor perturbations evolve independently at the linear order but are dynamically coupled at the second and higher orders [6]. The scalar mode related to the primordial density perturbations can excite the tensor mode, inducing secondary GWs when they are either localized at scales much smaller than the Hubble horizon, during inflation, or if they reenter the horizon during the post-inflationary, radiation-dominated, and matter-dominated epochs. This phenomenon can generate a relic GW stochastic background. As is well known, LSS and CMB observations constrain the primordial fluctuations to be distributed to an approximately Gaussian distribution at scales larger than 1 mpc; the power spectrum is nearly scale-invariant with an amplitude which is too small for current detection. On the other hand, the constraints are less stringent for scales much below the mpc, and a large enhancement of scalar and tensor perturbations can be possible [7]. Several possible mechanisms for an amplification of primordial density perturbations and induced GWs at small scale were proposed in the literature, including double-inflation [8,9] with PBH formation.

When we consider early universe origins for PBHs, one possibility is that it is related to inflation dynamics. In some cases, inflation(s) can trigger the formation of critical matter overdensities and thus it can catalyze the formation of PBHs. Typically, PBHs do not have astrophysical dimension and mass; moreover they may be copiously produced and

constitute the whole CDM. Surprisingly, MSG can provide an assist to the PBH scenario. First, MSG naturally relates Starobinsky's scalaron with other scalar field partners which can participate in the inflation dynamics. In other words, MSG establishes a natural (super)symmetric principle for multi-inflation. In the simplest MSG cases, a double-inflation scenario can be envisaged. It is well known that double-inflation can efficiently provide a PBH CDM genesis. However, most of scenarios proposed in the literature are benchmark models self-engineered in order to achieve the "right" PBH amount. On the other hand, MSG naturally realizes a double inflation scenario for an efficient generation of PBHs [8,9]. (Local) supersymmetry also dictates the structure of the two scalars' interactions which are compatible with PBH CDM from inflation in a large parametric space. Another possibility is that CDM is partially composed by PBHs and gravitinos produced during inflation, as discussed in [10–13].

Now we are ready to come to the interesting aspect of it: testability. Indeed, the power spectrum of scalar perturbations produced by double-inflation in turn originated by MSG not only generates PBH CDM but it also sources a GW stochastic background detectable within LISA (and similar design concepts) sensitivity (estimated) curve(s).

Thus, in our model, supersymmetry can be spontaneously broken at much higher energy than the TeV-scale. In this sense, SUSY can be interpreted as a symmetry principle for inflation and dark matter rather than related to the Higgs hierarchy problem or grand unification theories (GUTs). In such a case, SUSY may be searched in the sky from GWs rather than in colliders. MSG is also a natural candidate for generating PBHs and GWs from sound speed resonances [14–18]. Indeed, a sound speed oscillation around unity during inflation may occur in certain parametric regions around turning points in multi-inflaton trajectories.

Our paper is organized as follows. In Section 2 we will review the main feature of MSG model; in Section 3, we discuss PBH production; in Section 4 GW phenomenology related to PBH production is shown; in Section 5 we show sound speed resonance phenomena; in Section 6 our conclusions and remakrs.

## 2. Model

Let us consider a MSG model in superfield formalism as follows:

$$\mathcal{L} = \int d^2\Theta\, 2\mathcal{E}\left[ -\frac{1}{8}(\bar{\mathcal{D}}^2 - 8\mathcal{R})\mathcal{N}(\mathcal{R}, \bar{\mathcal{R}}) + \mathcal{F}(\mathcal{R}) \right] + h.c. \tag{1}$$

where $\mathcal{N}(\mathcal{R}.\bar{\mathcal{R}})$ and $\mathcal{F}(\mathcal{R})$ are arbitrary functions and $\mathcal{R}$ is the chiral scalar curvature superfield; $\mathcal{E}$ is the chiral density superfield, $\mathcal{D}_\alpha, \bar{\mathcal{D}}_{\dot\alpha}$ are the superspace covariant derivative with $\mathcal{D}^2 \equiv \mathcal{D}^\alpha\mathcal{D}_\alpha$ & $\bar{\mathcal{D}}^2 \equiv \bar{\mathcal{D}}_{\dot\alpha}\bar{\mathcal{D}}_{\dot\alpha}$, $\Theta$ are the superspace coordinates. We formulate MSG by using standard supergravity formalism in curved superspace (see [19]).

A reasonable power series ansatz is the following:

$$\mathcal{N} = \frac{12}{M^2}|\mathcal{R}|^2 - \frac{72}{M^4}\zeta|\mathcal{R}|^4 - \frac{768}{M^6}\gamma|\mathcal{R}|^6, \tag{2}$$

$$\mathcal{F} = -3\mathcal{R} + \frac{3\sqrt{6}}{M}\delta\mathcal{R}^2, \tag{3}$$

where $M$ is related to the scalaron mass, $\zeta, \gamma, \delta$ are free parameters. Einstein's supergravity is obtained in the limit of $\mathcal{N} \to 0$, $\mathcal{F} \to -3\mathcal{R}$, i.e., for $\zeta, \gamma, \delta \to 0$ the simplest extension of $R + R^2$ gravity is re-obtained.

The relevant bosonic sector in the Einstein frame [8,9] reads as

$$e^{-1}\mathcal{L} = \frac{1}{2}R - \frac{1}{2}(\partial\varphi)^2 - \frac{3M^2}{2}Be^{-\sqrt{2/3}\varphi}(\partial\sigma)^2 - \frac{1}{4B}\left(1 - Ae^{-\sqrt{2/3}\varphi}\right)^2 - e^{-2\sqrt{2/3}\varphi}U \tag{4}$$

where $A \equiv A(\sigma), B \equiv B(\sigma), U \equiv U(\sigma)$ are given by

$$A(\sigma) = 1 - \delta\sigma + \frac{1}{6}\sigma^2 - \frac{11}{24}\zeta\sigma^4 - \frac{29}{54}\gamma\sigma^6 \tag{5}$$

$$B(\sigma) = \frac{1}{3}M^{-2}(1 - \zeta\sigma^2 - \gamma\sigma^4),\tag{6}$$

$$U(\sigma) = \frac{1}{2}M^2\sigma^2\left(1 + \frac{1}{2}\delta\sigma - \frac{1}{6}\sigma^2 + \frac{3}{8}\zeta\sigma^4 + \frac{25}{54}\gamma\sigma^6\right).\tag{7}$$

Here, $e \equiv \det(e_m^a)$ (the scalar vielbien) is the first component of $\mathcal{E}$ in supersymmetric $\Theta$-expansion, $R$ is the Ricci scalar contained in $\mathcal{R}$ superfield, $\varphi$ is the Starobinsky scalar field and $\sigma$ is a new scalar field contained in both $\mathcal{R}$ and $\mathcal{E}$. In principle extra supersymmetric partners $b_m, a$ are contained in the superfields and propagating d.o.f in the theory; for simplicity we considered them as frozen around inflation dynamics.

The relevant field equations in FLRW are

$$0 = \ddot{\varphi} + 3H\dot{\varphi} + \frac{1}{\sqrt{6}}(1 - \zeta\sigma^2 - \gamma\sigma^4)e^{-\sqrt{\frac{2}{3}}}\dot{\sigma}^2 + \partial_\varphi V,\tag{8}$$

$$0 = \ddot{\sigma} + 3H\dot{\sigma} - \frac{\zeta\sigma + 2\gamma\sigma^3}{1 - \zeta\sigma^2 - \gamma\sigma^4}\dot{\sigma}^2 - \sqrt{\frac{2}{3}}\dot{\varphi}\dot{\sigma} + \frac{e^{\sqrt{\frac{2}{3}}\varphi}}{1 - \zeta\sigma^2 - \gamma\sigma^4}\partial_\sigma V\tag{9}$$

$$0 = \frac{1}{2}\dot{\varphi}^2 + \frac{1}{2}(1 - \zeta\sigma^2 - \gamma\sigma^4)e^{-\sqrt{\frac{2}{3}}\varphi}\dot{\sigma}^2 + \dot{H}\tag{10}$$

$$0 = V - 3H^2 - \dot{H},\tag{11}$$

$$V = \frac{1}{4B}\left(1 - Ae^{-\sqrt{\frac{2}{3}}\varphi}\right)^2 + e^{-2\sqrt{\frac{2}{3}}\varphi}U(\sigma),\tag{12}$$

where $H$ is the Hubble rate.

Such a system of two scalar fields not linearly coupled include the Starobinsky scalaron and it can lead to a double inflation dynamics.

## 3. Primordial Black Holes

In this section, we discuss the production of PBHs in modified supergravity, PBHs can be copiously produced in a large region of parametric space of Equation (1). To fix the ideas, let us distinguish the case of sharp and smooth peaks in the power spectrum; for example $\delta \simeq 0.6$ and $\delta \simeq 0.1$, respectively (where $\delta$ is the parameter introduced in Equation (3)).

Interestingly, we find that PBHs can account for the total amount of CDM as $f_{tot} \simeq 1$ in a parametric domain $1/3 \leq \delta \leq 2/3$. PBH mass can be estimated as in [20]:

$$M_{PBH} \simeq \frac{M_{Pl}^2}{H(t_*)}\exp\left[2(N_{end} - N_*) + \int_{t_*}^{t_{exit}}\epsilon(t)H(t)dt\right],\tag{13}$$

where $M_{PBH}$ is the PBH mass, $H(t)$ is the Hubble rate at cosmological time $t$, $N_{end}, N_*$ are the e-fold numbers at the end of double inflation and at $t_*$ respectively, $t_*$ is the time when the first slow-roll ends, $t_{exit}$ is the time when the CMB pivot scale $k$ exits the horizon. The PBH mass depends on the slow-roll parameter $\epsilon \equiv -\dot{H}/H^2$. Therefore it is also related to inflationary parameters $n_s(\epsilon), r(\epsilon)$ which have to be compatible, at least at $3\sigma$, with Planck data $n_s = 0.9649 \pm 0.0042$ ($1\sigma$ C.L.) and $r < 0.064$ ($2\sigma$ C.L.) [21].

PBH mass $\tilde{M}_{PBH}(k)$, production rate $\beta_f(k)$ and density contrast coarse-grained $\sigma^2(k)$ spectra can be estimated as in [22,23],

$$\tilde{M}_{PBH} \simeq 10^{20}\left(\frac{7 \times 10^{12}}{k\,\mathrm{Mpc}}\right)^2\,\mathrm{g},\quad \beta_f(k) \simeq \frac{\sigma(k)}{\sqrt{2\pi}\delta_c}e^{-\delta_c^2/2\sigma^2 k},\tag{14}$$

$$\sigma^2(k) = \frac{16}{81}\int\frac{dq}{q}\left(\frac{q}{k}\right)^4 e^{-q^2/k^2}P_\zeta(q),\tag{15}$$

while the PBH/CDM ratio is

$$f(k) \equiv \frac{\Omega_{PBH}(k)}{\Omega_{DM}} \simeq \frac{1.2 \times 10^{24} \beta_f(k)}{\sqrt{\tilde{M}_{PBH}(k)g^{-1}}} . \tag{16}$$

In Table 1 we summarize results. For $\delta = 0.09, 0.61$, we can consider $\Delta N_2 \leq 19, 20$ respectively, where $\Delta N_2$ is the second inflation slow-roll e-fold difference number. We obtain an almost monochromatic PBH mass distribution with $M_{PBH} \sim 10^{20}, 10^{18}$ g respectively. Moreover, the primordial scalar power spectra as well as the abundance of the PBHs of three representative I, II and III cases parameterized by Table 2 are shown in Figures 1 and 2, respectively. Such a result is compatible with all multi-messenger constraints.

**Table 1.** PBH masses in cases of $\delta = 0.09$ and $\delta = 0.61$. We compare the PBH mass in grams $M_{PBH}$ with e-fold number $\Delta N_2$ of the second inflation stage, the scalar tilt parameter $n_s$ and the maximal scalar to tensor ratio $r_{max}$ predicted in our model at the CMB scale.

|  | $\delta = 0.09$ | | | | $\delta = 0.61$ | | | |
|---|---|---|---|---|---|---|---|---|
| $\Delta N_2$ | 10 | 17 | 20 | 23 | 10 | 17 | 20 | 23 |
| $M_{PBH}(g)$ | $10^9$ | $10^{15}$ | $10^{18}$ | $10^{20}$ | $10^9$ | $10^{15}$ | $10^{18}$ | $10^{20}$ |
| $n_s$ | 0.9566 | 0.9486 | 0.9443 | 0.9390 | 0.9581 | 0.9504 | 0.9461 | 0.9409 |
| $r_{max}$ | 0.005 | 0.007 | 0.008 | 0.010 | 0.004 | 0.006 | 0.007 | 0.008 |

**Table 2.** A table of model parameters with three interesting cases for PBH CDM is shown; $\gamma, \delta$ are the MSG lagrangian parameters, $\Delta N_2$ the e-fold number of second stage of inflation, $n_s, r$ the scalar tilt parameter and the scalar to tensor ratio respectively, $\delta_c$ the critical density.

|  | $\gamma$ | $\delta$ | $\Delta N_2$ | $\delta_c$ | $n_s$ | $r$ |
|---|---|---|---|---|---|---|
| Case I | 1.5 | 0 | 20 | 0.4 | 0.942 | 0.009 |
| Case II | 0 | 0.09 | 19 | 0.47 | 0.946 | 0.008 |
| Case III | 0 | 0.61 | 20 | 0.4 | 0.946 | 0.007 |

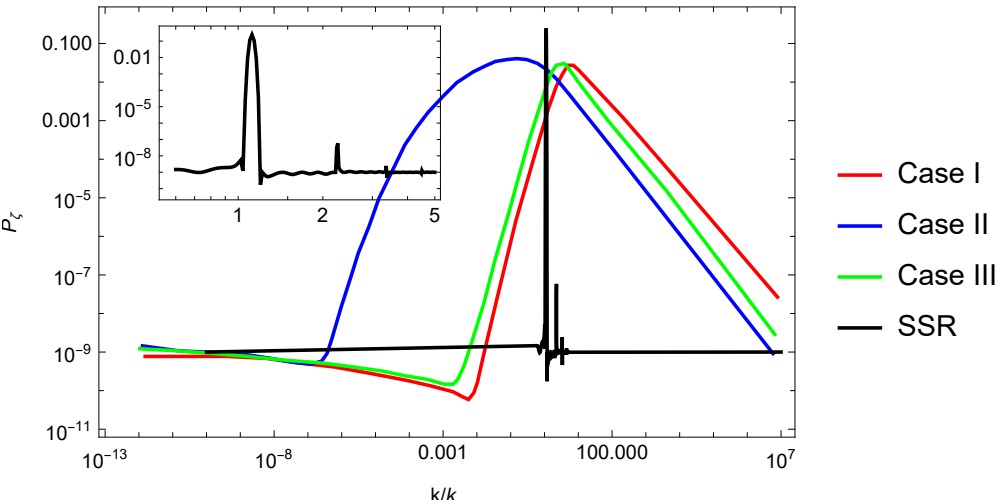

**Figure 1.** The power spectra of scalar perturbations of three representative cases for PBH CDM.

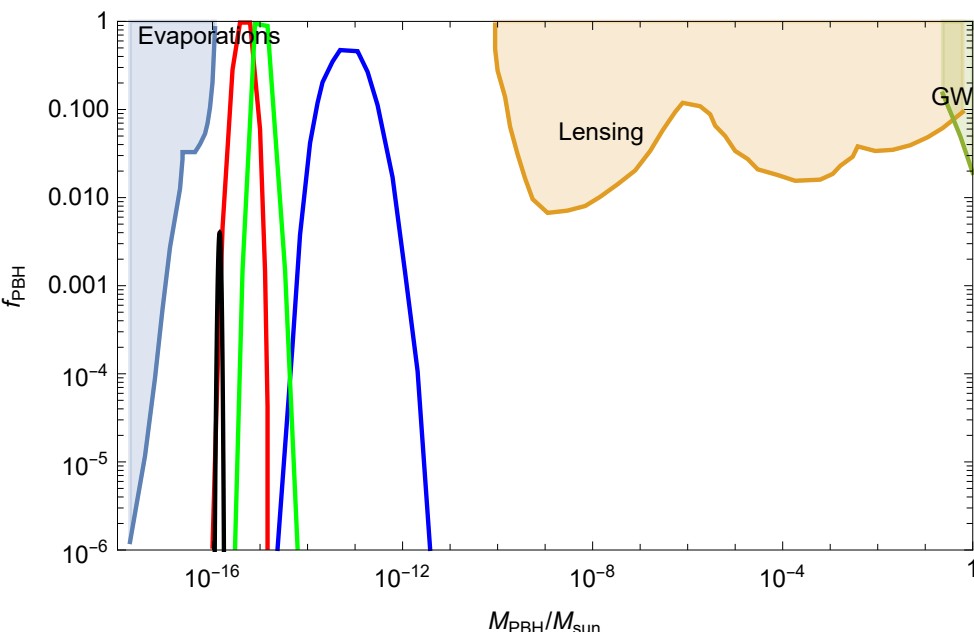

**Figure 2.** The respective PBH density fractions to three power perturbation spectra in Figure 1. A comparison with experimental constraints is displayed.

## 4. Energy Density of Induced Gravitational Waves

The generation of a high peak power spectrum of scalar perturbations not only catalyzes the formation of PBHs but it also sources GW stochastic background appearing as a relic signal around mHZ frequencies today. The GW density in present universe is given by [24]

$$\frac{\Omega_{GW}(k)}{\Omega_r} = \frac{c_g}{72} \int_{-1/\sqrt{3}}^{1/\sqrt{3}} dd \int_{1/\sqrt{3}}^{\infty} ds \left[ \frac{(s^2 - \frac{1}{3})(d^2 - \frac{1}{3})}{s^2 + d^2} \right]^2 P_\zeta(kx) P_\zeta(ky)(I_c^2 + I_s^2), \tag{17}$$

where in SM the constant $c_g \simeq 0.4$ and $c_g \simeq 0.3$ in minimal supersymmetric standard model (MSSM). The present radiation density $h^2 \Omega_r \simeq 2.47 \times 10^{-5}$ from CMB data and $h$ is the reduced Hubble parameter.

In Equation (17), we have defined two integration variables as functions of $s, d$;

$$x, y = \frac{\sqrt{3}}{2}(s \pm d) \tag{18}$$

and $I_c$, $I_s$ have the following integral forms:

$$I_c = -8 \int_0^\infty d\eta \, \sin \eta \{ [T(x\eta + x\eta T'(x\eta)][T(y\eta + x\eta T'(y\eta)]T(x\eta)T(y\eta) \}, \tag{19}$$

$$I_s = 8 \int_0^\infty d\eta \, \cos \eta \{ [T(x\eta + x\eta T'(x\eta)][T(y\eta + x\eta T'(y\eta)]T(x\eta)T(y\eta) \}, \tag{20}$$

where

$$T(k\eta) = \frac{9}{(k\eta)^2} \left[ \frac{\sqrt{3}}{k\eta} \sin\left(\frac{k\eta}{\sqrt{3}}\right) - \cos\left(\frac{k\eta}{\sqrt{3}}\right) \right], \tag{21}$$

and $\eta$ is the conformal time.

From analytic integration one obtains

$$I_c = -36\pi \frac{(s^2 + d^2 - 2)^2}{(s^2 - d^2)^3} \theta(s - 1), \tag{22}$$

$$I_s = -36 \frac{s^2 + d^2 - 2}{(s^2 - d^2)^2} \left[ \frac{s^2 + d^2 - 2}{s^2 - d^2} \log \left| \frac{d^2 - 1}{s^2 - 1} \right| + 2 \right].$$ (23)

Thus, from these formulas we can compute the GW spectrum induced by scalar perturbations. Theoretical results are compared with sensitivity curves of future experiments and the situation is summarized in Figure 3. It is clear that the GW signal is detectable from LISA, TianQin, TAIJI and DECIGO.

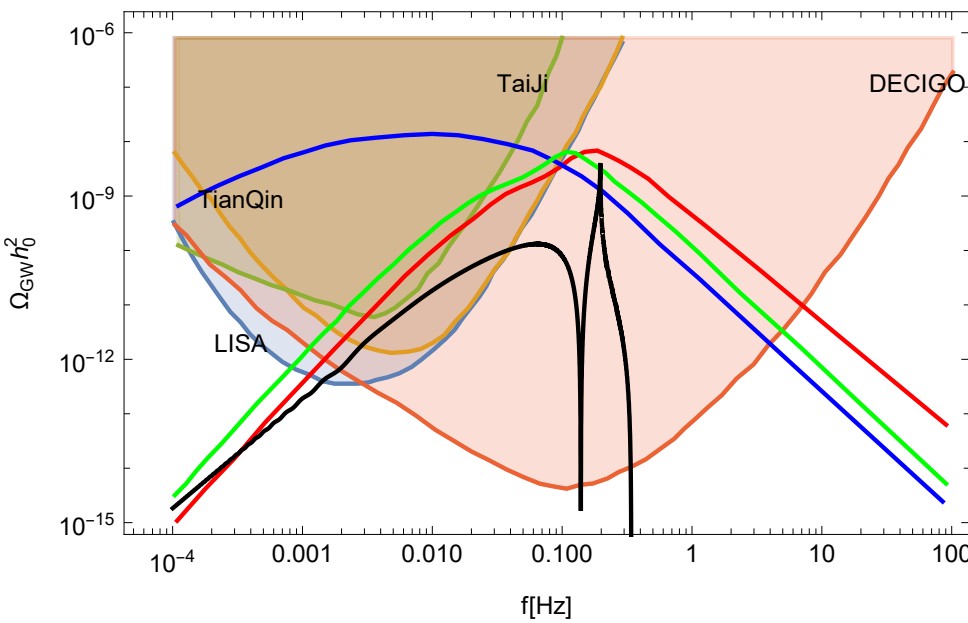

**Figure 3.** A comparison between stochastic GW density spectrum induced in our supergravity models and expected sensitivity curves of space-based experiments.

## 5. Sound Speed Resonance (SSR)

The two-field inflation model in the MSG provides a natural realization for relevant fields with varying sound speed. The nontrivial sound speed of the inflaton or curvaton in single/multiple field inflation paradigms are widely explored in the literature [14–17]. In particular, in [15] the authors assumed an oscillatory modification to the sound speed

$$c_s^2 = 1 - 2\xi(1 - \cos(2k_* \tau))$$ (24)

where $c_s$ is the sound speed, $\xi, k_*$ are characteristic constants, and $\tau$ is the cosmological conformal time. From a phenomenological point of view, it can be demonstrated that the primordial power spectrum can be dramatically amplified in a narrow band characterized by $k_*$ due to the parametric resonance effect (see the black line in Figure 1). Consequently, the narrow peak of the density spectrum enhances the abundances of the PBH formation as well as the induced GWs in a different manner compared to the broad spectra in Cases I, II, and III (see Figures 2 and 3).

On the other hand, the inflaton oscillates at the turning point of the trajectory in the two-field target space, which is usually accompanied with the oscillation of its sound speed [16,17]. Therefore, the MSG framework provides a theoretical origin of the sound speed resonance mechanism, leading to the significant enhancement of power spectrum of the curvature perturbations as well as the induced GWs. Moreover, it is worth mentioning that the amplification of the primordial power spectrum caused by a sudden turn of the inflaton trajectory was observed in [14].

## 6. Conclusions and Remarks

In this paper, we discussed the MSG phenomenology from GWs and PBH genesis. We consider MSG embedding Starobinsky's $R^2$-gravity in superfield formalism. In Einstein's

frame, MSG effectively reduces to EH action with two scalar partners non-linearly coupled participating to (double) inflation dynamics. The structure of the interactions among the two scalars is rigidly dictated by (local) supersymmetry. We showed that MSG elegantly predicts PBH genesis as an explanation of CDM. Such a model can be tested from GWs with LISA, DECIGO, BBO, TianQin, and TAIJI projects. We also put forward the idea of sound speed resonances during inflation from MSG, which we think it deserves for future deeper investigations as an alternative mechanisms for PBH genesis. (See also Ref. [25] for several updates on LISA cosmology working group).

To summarize, MSG appears to be an attractive unified theory for inflation and CDM. On the other hand, it is natural to extend Starobinsky's supergravity toward a more general supergravity version of $f(R)$-theory. In this sense, also late universe acceleration from (dynamic) dark energy would be included in a unified scheme for the dark side [26].

Considering GW and multi-messenger physics, we are also tempted to suggest the existence of MSG star solution beyond general relativity, a supergravastar. In [27], in collaboration with S. Ketov, one of us showed that MSG in general also deforms energy conditions. Thus, it is natural to consider modifications of the equation of state for neutron stars; as well exotic compact objects (ECOs) [28–30] or dynamic horizon solutions [31,32] are not precluded possibilities. These are particular exciting in multi-messenger era searching for deviations from GR predictions in black holes or neutron star mergings. Recently, other results in modified gravity and GWs also encourage us to explore the multi-messenger phenomenology of their supergravity extensions [33–35].

To conclude, MSG phenomenology appears to be a vibrant new research direction at the dawn of the multi-messenger era. Tests of supergravity in cosmology is fundamental for our understanding of quantum gravity which in turn continues to remain elusive to us.

**Author Contributions:** We equally contribute to all parts of this work. All authors have read and agreed to the published version of the manuscript.

**Funding:** Talent Scientific Research Program of College of Physics, Sichuan University, Grant No. 1082204112427; Fostering Program in Disciplines Possessing Novel Features for Natural Science of Sichuan University, Grant No. 2020SCUNL209; 1000 Talent program of Sichuan province 2021; scholarship from China Scholarship Council (CSC) under the Grant CSC No. 202106240085.

**Institutional Review Board Statement:** Not applicable.

**Informed Consent Statement:** Not applicable.

**Data Availability Statement:** Not applicable.

**Acknowledgments:** A.A. would like to thank Yermek Aldabergenov and Sergei Ketov for several discussions and cooperations on these subjects. A.A. work is supported by the Talent Scientific Research Program of College of Physics, Sichuan University, Grant No. 1082204112427 and the Fostering Program in Disciplines Possessing Novel Features for Natural Science of Sichuan University, Grant No. 2020SCUNL209 and 1000 Talent program of Sichuan province 2021. Q.G. work is supported by the scholarship from China Scholarship Council (CSC) under the Grant CSC No. 202106240085.

**Conflicts of Interest:** The authors declare no conflict of interest.

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
