# Peer review of "Modified Supergravity Phenomenology in Gravitational Waves Era"

_universe, doi:10.3390/universe8050280_

Round 1

Reviewer 1 Report

The authors performed an interesting study related with the stochastic gravitational wave background generated by supergravity theories. I have several questions:

  1. The authors state in several places and even in the abstract that in the context of their theories, a large amplification of the scalar perturbation power spectrum is generating a GW stochastic background occurs. I dont understand that statement. The future interferometer experiments will directly probe a stochastic gravitational wave background which is generated by adiabatical streched primordial tensor perturbations, which in turn correspond to tensor modes which became subhorizon during the first stages of reheating. Thus we are talking about tensor modes, not scalar perturbations. Scalar perturbations become non-linear for modes with weavelengths lower than 10Mpc, which essentially cover all the modes with frequencies probed by the Nanograv and up to the LISA DECIGO ET and BBO experiments. So perhaps the authors mean tensor perturbations? This must be corrected. Scalar perturbations cannot be probed by future interferometers for sure, the scalar perturbations have CMB frequencies and the tensor perturbation background in scalar modes of CMB can be traced indirectly via the B-mode pattern corresponding to low multipoles of the CMB or large angular scales. The authors must clarify and correct this important issue.
  2. Some recent literature is missing, regarding modified gravity effects on primordial gravitational waves, for example 2204.06304,2204.05434, 2204.00876, 2203.10599.
  3. What is the motivation for using supergravity theories? Supersymmetry seems to be absent from high energy physics, so what is the motivation to use a supersymmetric context? Furthermore, how does supersymmetry scale affects the energy spectrum of the primordial gravitational waves? Is there a frequency range in which the peak is affects by the susy breaking or susy scale?

After the above comments are addressed, the article can be accepted for publication.

Author Response

Dear editor, 

We wish to resubmit our report titled “Modified Supergravity phenomenology in Gravitational waves era” to the 

Special issue titled “Dark Energy and the Dark Sector in Supergravity, String Theory and Extra-Dimensions”. 

We would like to thank the anonymous referee of her/his constructive criticism and comments. 

We wish to systematically reply to the 3 referee questions as follows. 

  1. We rephrased several comments in abstract and introduction since we agree that in previous version they may sound not clear. In our case, the double-inflation dynamics amplify the scalar perturbation power spectrum at distance <<Mpc. At linear order, scalar and tensor modes are decoupled but at non-linear orders they are dynamically coupled. Thus the scalar modes can excite tensor modes inducing secondary GWs in form of a stochastic background. As remarked by the referee, scalar perturbations cannot be directly detected in space-based interferometers or in Pulsar timing radio-astronomy. Our testable prediction is from tensor mode perturbations detachable as GWs as shown in Fig.5. 
  2. We thank the referee for these recent references that we added in our new version.
  3. In our case, the supergravity extension of Modified gravity is though as a “symmetry principle” dictating a rigid structure of Lagrangian interactions between the Starobinsky scalaron and a new scalar field. In other words, supersymmetry can be a “guidance” towards the formulation of a multi-inflation scenario and PBH dark matter: masses, couplings and interaction terms are not arbitrary or self-engineered ad hoc but they are consistent with supersymmetry algebra. Indeed, in Starobinsky’s supergravity, the mass of the Starobinsky scalaron and the new scalar field are interrelated by supersymmetry. Such a picture reverses the common TeV supersymmetry paradigm which we briefly summarize as follows. Supersymmetry was invoked as an explanation of the Higgs hierarchy problem; postulating it as spontaneously or dynamically broken at the TeV energy scale. Negative results of LHC indicate that TeV-supersymmetry and its relation with the Higgs boson physics are excluded. A minimal supersymmetric standard model (MSSM) also predicts several possible candidates of Weak Interacting Massive Particles (WIMPs) as the neutralino, not far from electroweak to TeV-scale. Also in these directions, negative results in Direct and Indirect dark matter searches suggest that TeV-supersymmetry is probably not related to cold dark matter. In our scenario, supersymmetry cannot be spontaneously broken at TeV-scale; it has to be broken at Starobinsky’s inflation scale. In other words, we are not exploring the conventional supersymmetric extensions of the Standard Model as an attempt to stabilize the electroweak scale. On the contrary, we consider the possibility that supersymmetry is related to the inflation scale and thus it is spontaneously broken at much higher scale than TeV. In our case, dark matter is not obtained as WIMPs at all but it is naturally obtained in form of PBHs, in turn generated by double-inflation. The PBH mass spectrum is particularly sensitive to the double inflation potential, in turn controlled in structure by supersymmetry and few free parameters. The GW spectrum and peak is also sensitive to the double inflation potential from supergravity. If supersymmetry is not broken at the Starobinsky mass scale, then the mechanism cannot work. Thus we find that high scale supersymmetry can be tested in cosmology from GWs rather than in colliders. On the other hand, theoretical searches of a UV completion of quantum gravity motivate exploration of supergravity ad a key element of superstring theory and its impact In early universe physics. 

Best regards,

The authors

Reviewer 2 Report

This communication attempts to briefly describe and summarise a range of phenomenological aspects of modified supergravity. This naturally touches on many subject areas: including early Universe cosmology and inflation, dark matter and primordial black holes, gravitational wave astronomy across a range of frequencies, and theoretical gravity. Therefore, this paper should be understandable to a broad audience. Unfortunately, this is not the case, and I must ask the authors to make significant changes to the manuscript before I can give a meaningful review.

To keep this discussion brief, I will focus just on the equations in section 2. In these few equations the authors introduce numerous quantities without any definitions or explanation: including \Theta, E, D, e, \sigma, \varphi, b_m, a, H, V, and the overbar. Many of these may be standard notation in particular fields, and I’m sure many are defined in the papers cited; however, notations and conventions vary and the authors must make an effort to at least briefly introduce and define them here, and ideally provide some background information as well. Further on in the paper the quantity r_max is also mentioned and plays a major role in the results (see Table 1) but this quantity is also not defined (and there are more examples in later sections of the paper). Addressing these issues will require adding quite a bit of text at various places in the papers. Fortunately, the paper is currently quite short and there is space to do this.

In its present form, I find this paper to be impossible to read and understand for the reasons given above. Therefore, I have not made any attempt to review the scientific content of the paper. I encourage the authors to carefully rewrite the paper with everything carefully defined and adding lots more additional background information with the aim of helping the general reader to follow the discussion. The subject matter is interesting, and I would be happy to read and review again a revised version of this manuscript.

Author Response

Dear editor, 

We wish to resubmit our report titled “Modified Supergravity phenomenology in Gravitational waves era” to the 

Special issue titled “Dark Energy and the Dark Sector in Supergravity, String Theory and Extra-Dimensions”. 

We would like to thank the anonymous referee of her/his constructive criticism and comments. 

In the second version of our paper, we worked to improve the exposition of our report which, as rightly remarked by the referee, touches many different subject areas. 

First of all, we work to define all variables which appear in section 2-3, tables and plot. 

Following the suggestions of the referee, we also added more definitions and explanations on supergravity formalism as well as in cosmological inflation parameters. 

Best regards, 

The authors 

Reviewer 3 Report

In the submitted manuscript, the authors have studied phenomenological aspects of Modified Supergravity, in particular gravitational waves and generation of primordial black holes. The manuscript is well-structured and clearly presented, and it will be of interest to a wide range of audience working on astronomy, gravitation and cosmology. I recommend it to be published in its current form.

Author Response

Dear editors, 

the third referee accepted our manuscript. 

best regards, 

the authors

Round 2

Reviewer 1 Report

The authors performed considerable revision, the article can be accepted for publication.

Reviewer 2 Report

The authors have made changes to address the concerns raised in my first report.